# Imbalanced Multi-instance Multi-label Learning via Coding Ensemble and Adaptive Thresholds

Xinyue Zhang
National University of Defense Technology
Changsha, China
zhang0331zxy@163.com

Tingjin Luo*
National University of Defense Technology
Changsha, China
tingjinluo@hotmail.com

Yueying Liu
National University of Defense Technology
Changsha, China
liuyueyingnudt@163.com

Chenping Hou
National University of Defense Technology
Changsha, China
hcpnudt@hotmail.com

## Abstract

Multi-instance multi-label learning (MIML), which deals with objects with complex structures and multiple semantics, plays a crucial role in various fields. In practice, the naturally skewed label distribution and label dependence contribute to the issue of label imbalance in MIML, which is crucial but rarely studied. Most existing MIML methods often produce biased models due to the ignorance of inter-class variations in imbalanced data. To address this issue, we propose a novel imbalanced multi-instance multi-label learning method named IMIMLC, based on the error-correcting coding ensemble and an adaptive threshold strategy. Specifically, we design a feature embedding method to extract the structural information of each object via Fisher vectors and eliminate inexact supervision. Subsequently, to alleviate the disturbance caused by the imbalanced distribution, a novel ensemble model is constructed by concatenating the error-correcting codes of randomly selected subtasks. Meanwhile, IMIMLC trains binary base classifiers on small-scale data blocks partitioned by our codes to enhance their diversity and then learns more reliable results to improve model robustness for the imbalance issue. Furthermore, IMIMLC adaptively learns thresholds for each individual label by margin maximization, preventing inaccurate predictions caused by the semantic discrepancy across many labels and their unbalanced ratios. Finally, extensive experimental results on various datasets validate the effectiveness of IMIMLC against state-of-the-art approaches.

## CCS Concepts

• **Computing methodologies** → **Semi-supervised learning settings**; *Ensemble methods*.

---

*Corresponding author

## Keywords

Multi-instance Multi-label Learning, Imbalanced Data, Error Correcting Output Code, Feature Mapping

**ACM Reference Format:**
Xinyue Zhang, Tingjin Luo, Yueying Liu, and Chenping Hou. 2024. Imbalanced Multi-instance Multi-label Learning via Coding Ensemble and Adaptive Thresholds. In *Proceedings of the 32nd ACM International Conference on Multimedia (MM '24), October 28-November 1, 2024, Melbourne, VIC, Australia.* ACM, New York, NY, USA, 10 pages. https://doi.org/10.1145/3664647.3680911

## 1 Introduction

A complicated object in many practical applications contains several labels simultaneously and its own intrinsic structure, which may be thought of as a bag of instances. Multi-instance multi-label learning (MIML) is a weakly supervised learning paradigm [45] that can handle such samples. By simulating the interaction between instances and the label set, it enhances the generalization capacity of the model by helping it comprehend the multiple meanings and overall structure of the object. With adaptation to different practical scenarios by adjusting the combination of instances, MIML is highly flexible and has been applied to various fields, such as text analysis [4, 42], audio and video location mining [11, 39, 40], signal recognition [24], and medical diagnosis [15, 21], etc.

In practice, it is typical for the label distribution to be inherently skewed, which inevitably raises the problem of label imbalance. For example, in breast medical imaging diagnosis, a single medical image viewed as a bag consists of many tissue pixels, as shown in Fig. 1. Because cancer patients make up a small percentage of the population overall, there are often one to several orders of magnitude fewer pixels of tumor tissue than of normal tissue. As a result, it has a significant label imbalance issue due to bags and disease-related cases being minority samples, which is a typical long-tailed distribution. In addition, label dependence will exacerbate the issue of imbalance. On the one hand, various bags usually match various numbers of label classes. As an illustration, Fig. 1 (a) shows three different kinds of lesions in the breast tissue. Actually, there is only one label for "normal" in an image of normal tissue, while others with more severe and intricate pathological processes include several breast tissue lesion labels. This leads to an increase in the difference in the number of samples that each label class possesses. On the other hand, certain labels have a greater co-occurrence frequency due to label correlation. For example, pathological processes

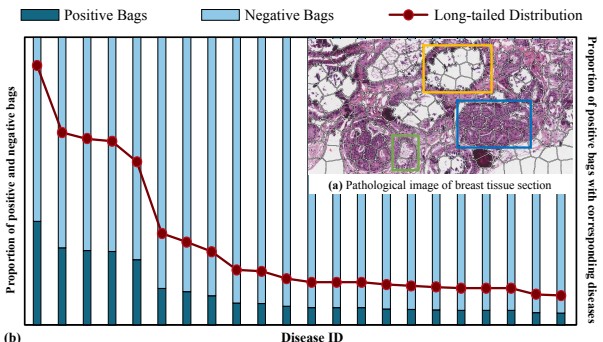

(a) Pathological image of breast tissue section

**Figure 1: An example of the label imbalance issue in the breast medical imaging diagnosis task. (a) different colored rectangular boxes mark different breast tissue lesion labels. (b) presents the long-tailed distribution of label classes.**

with the same causes often co-occur. Concurrent identification of related lesions can proceed simultaneously when the image contains specific lesion labels. As the number of samples with certain labels increases, there are more samples with related labels, making the problem of imbalance more severe and difficult to solve.

Compared with traditional classification tasks, there are two main challenges in handling such imbalanced MIML problems. (1) In MIML tasks, only bag-level labels are available, while labels of instances recording features are unknown. The inconsistency of levels at which features and labels are located makes the task very complicated. (2) In imbalanced distributions, it is more difficult to handle tasks due to the effect of class overlap, data selection bias, and insufficient prior being magnified.

To solve these problems, many algorithms have been proposed in the literature. To address coarse-grained labeling issues, there are many MIML methods. Degradation methods transform the original MIML task into either a multi-instance learning (MIL) task or a multi-label learning (MLL) task to eliminate inexact supervision. Some work discretizes bag-level labels to label instances to train an instance-level classifier [48], and some work represents the bag as a new feature vector using the information provided by instance features to learn a bag-level classifier [16, 18, 37, 42, 48]. In recent years, some work has jointly learned the bag-level classifier and instance-level classifier to cover both features and labels simultaneously [35, 41]. To solve label imbalance problems, there are many imbalanced learning methods. Resampling methods generate or throw some samples to balance data distribution [2, 8, 30, 38]. Reweighting methods assign different weights to samples to balance the importance of different sizes of the population for the model [5, 17, 22, 44]. Ensemble methods incorporate multiple base models to enhance recognition abilities for minority classes [3, 19, 26]. Meta-learning-based methods reweight samples through self-adaptation to tackle majority classes and minority classes differently [6, 7, 20, 23].

Although these approaches have been successfully applied to various applications, they still face several obstacles when directly used in the imbalanced MIML task. (1) Traditional MIML or imbalanced learning methods only focus on one aspect of our setting and are unable to effectively address our problems. (2) Due to the assumption of balanced label distributions, existing MIML algorithms

tend to focus on majority-class samples and ignore the contribution of minority classes, which results in biased models and decreased performance. (3) Traditional methods often assign labels to multi-semantic objects using a fixed threshold, resulting in unreliable predictions due to ignoring the semantic discrepancy among multiple labels and the imbalanced ratios of different classes.

To solve these problems in this new learning scenario, we propose an imbalanced multi-instance multi-label learning method (IMIMLC), based on the error-correcting coding ensemble and an adaptive threshold strategy, whose main framework is presented in Fig. 2. Specifically, we conduct feature embedding to extract the structural information of each object via the fisher vector and eliminate the effect of inexact supervision. Subsequently, to alleviate the disturbance of the imbalanced distribution, a novel ensemble strategy is constructed based on concatenating the error-correcting codes of randomly selected subtasks. IMIMLC trains multiple binary base classifiers on small-scale data blocks and learns the more reliable prediction to enhance the diversity of base learners and model robustness for imbalanced data. To avoid unreliable predictions caused by traditional fixed thresholds, we transform the threshold determination problem into learning classifiers with the maximum margin of soft labels, thereby IMIMLC adaptively learning threshold for each label. Finally, extensive experimental results on various MIML datasets verify the effectiveness of IMIMLC against state-of-the-art MIML approaches in terms of imbalance-specific evaluation metrics. In summary, the main contributions are listed as follows.

- We propose IMIMLC to solve a crucial but rarely studied problem, the label imbalance of MIML. As far as we know, this is the first attempt to tackle MIML tasks with the imbalanced label distribution.
- IMIMLC establishes a novel ensemble model and trains multiple base classifiers on randomly partitioned small-scale data blocks, which enhances the diversity of base learners and model robustness for the imbalanced distribution issue.
- To solve unreliable predictions caused by the ignorance of the discrepancy of semantics and class priors when using traditional manually fixed thresholds, IMIMLC adaptively learns thresholds for individual labels by maximizing the classification margin.
- Extensive experimental results on various datasets indicate that our IMIMLC outperforms other comparison methods in most cases and demonstrate its superiority and effectiveness.

## 2 Related Work

### 2.1 Overview of MIML

In practice, there are many MIML tasks. For example, in image recognition, a complete image is considered a bag, with its pixels as instances, and each image has one or several labels. Extracting information from pixels and predicting the labels of the image is a typical MIML task. The goal of MIML is to explore the relationship between an object described by multiple instances and a set of labels. Given $\{(\mathbf{B}_1, \mathbf{Y}_1), (\mathbf{B}_2, \mathbf{Y}_2), \ldots, (\mathbf{B}_N, \mathbf{Y}_N)\}$, where $\mathbf{B}_i$ represents a bag composed of a group of instances $\{x_i^1, x_i^2, \ldots, x_i^{n_i}\}$ and its label vector $\mathbf{Y}_i = [y_i^1, y_i^2, \ldots, y_i^C]$, where $y_i^l = 1$ when the $l$-th label is tagged for $\mathbf{B}_i$, and 0 otherwise. MIML learns $f_{\mathrm{MIML}} : 2^X \rightarrow 2^{\mathcal{Y}}$.

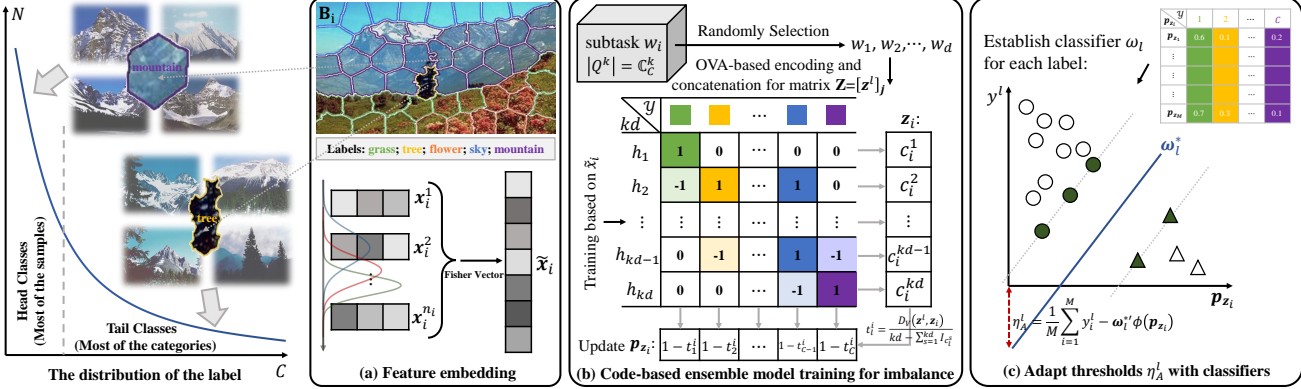

**Figure 2: Task scenario and the framework of IMIMLC. (a) Extract structural information from instances by feature embedding. (b) Concatenate OVA-based codes of subtasks to obtain the encoding matrix Z, based on which train ensemble model for imbalance. (c) Adapt thresholds with classifiers in the input space composed of feature vectors with semantics.**

Existing MIML methods proposed in the literature can be summarized according to the types of learned classifiers. With the assumption that each instance in a bag contributes equivalently and labels are independent, Zhang et al. [48] have proposed MIML-Boost and MIMLSVMmi algorithms by splitting multi-label sets into several binary classification label pairs to learn the instance-level classifier. To learn the bag-level classifier, Zhang and Zhou [48] have proposed the MIMLSVM algorithm that uses cluster to map for converting MIML samples into MLL samples. Based on this work, an increasing number of algorithms have been proposed for more robust classifiers or more accurate distance measurements, which achieved better performance [18, 37, 48]. Considering highly correlated labels often share information, Li et al. [16] have proposed the KISAR algorithm, which explores the reasonable relationship between input patterns and output labels to identify key instances that trigger labels. Similar to this work, the MIMLwel algorithm [42] tackles the weak labeling problem in MIML tasks where only a part of positive labels are tagged. In recent years, some work has considered learning at both the instance level and the bag level to train a joint classifier, which enriches the algorithms used for handling MIML tasks. Based on active learning, the CM2AL algorithm [35] queries the most probably positive instance-label pairs, labeling the bag based on these feedbacks. Founded upon attention mechanisms and manifold learning, the MIML-LLMC algorithm [41] handles bag structures and learns instance-label relations simultaneously.

## 2.2 Imbalanced Learning

In real MIML tasks, imbalance is a common characteristic of datasets. It can be viewed from three perspectives: imbalance within labels, imbalance between labels, and imbalance among the label sets [36]. The learning task with the label imbalance issue is more challenging. Specifically, during training, the model tends to concentrate more on the features and patterns of majority samples while ignoring important information from minority samples. It results in a weak ability of the model to recognize minority classes. Due to such insufficient learning, the model may mistakenly predict minority-class samples as majority-class samples, resulting in a lower recall

for minority classes. To solve these problems, an increasing number of methods have been proposed to adjust label distribution and enhance the ability of the model to recognize minority classes, thereby improving its performance and robustness.

**Resampling:** Resampling methods balance data distribution in the training set by over/undersampling [2, 8, 30, 38], or by filtering out noise through cleaning resampling [9, 13]. **Reweighting:** Reweighting methods assign different weights to samples to balance the importance of different sizes of the population for the model [5, 17, 22, 44]. **Ensemble Methods:** Ensemble methods improve recognition abilities for minority samples by combining multiple specific models [3, 26]. Liu et al. [19] have investigated the combination of the resampling method and the ensemble learning framework, which is proven effective. **Meta-learning Methods:** Meta-learning, which can also be formulated as a domain-adaptive strategy [10], has been applied to imbalanced learning [6, 7, 23, 25]. It tackles majority samples and minority samples differently and learns how to reweight through self-adaptation [33]. Liu et al. [20] have proposed the MESA algorithm, which learns the model from the imbalanced distribution based on the meta-sampler boost ensemble that trains the meta-sampler on task-agnostic meta-data and directly learns sampling strategies from the data.

## 3 Method

To tackle MIML tasks with imbalanced label distributions, we propose a novel imbalanced multi-instance multi-label learning method (IMIMLC) based on the error-correcting coding ensemble and an adaptive threshold strategy. Our IMIMLC mainly consists of feature embedding, code-based ensemble training, and classification threshold adapting, which are shown in Fig. 2. Specifically, IMIMLC first extracts structural information from instances and represents bags via fisher vectors to train the ensemble model on randomly partitioned data blocks. The ensemble model updates the bag representation with semantic annotation, based on which IMIMLC self-adapts classification thresholds for imbalanced label distribution. Besides, its computational complexity analysis is provided.

## 3.1 Efficient Bag Embedding Method

In MIML samples, only bag labels are available, while labels of instances that record features are unknown. To eliminate the effect of such inexact supervision, we extract the structural information of each bag by feature embedding.

Assume instances $\{x_i^1, x_i^2, \cdots, x_i^{n_i}\}$ in $\mathbf{B}_i$ are independently and identically distributed and generated from the Gaussian mixture model (GMM) $p$ consisting of $G$ components with parameter $\theta$, i.e., $p(x_i^j|\theta) = \sum_{g=1}^{G} \alpha_g p_g(x_i^j|\theta)$, where $\alpha_g \geq 0$ is the non-negative weight and satisfy the constraint $\sum_{g=1}^{G} \alpha_g = 1$, and $p_g(x_i^j|\theta)$ is the $g$-th Gaussian model. To retain distribution characteristics of instances and remove redundant information, the bag representation $\tilde{x}_i$ can be described as the fisher vector, which whitens the gradient of the log-likelihood of the GMM $p$ [31], i.e.,

$$\tilde{x}_i = \mathbf{L}_\theta \nabla_\theta \log p(\mathbf{B}_i|\theta) = \mathbf{L}_\theta \nabla_\theta \log p(x_i^1, x_i^2, \cdots, x_i^{n_i}|\theta)$$
$$= \mathbf{L}_\theta \nabla_\theta \log \prod_{j=1}^{n_i} p(x_i^j|\theta) = \mathbf{L}_\theta \sum_{j=1}^{n_i} \nabla_\theta \log p(x_i^j|\theta), \qquad (1)$$

where $\mathbf{L}_\theta$ is the Cholesky decomposition for whitening. Besides, similar to [31], we adopt the same normalization approach for $\tilde{x}_i$ to reduce variance dependence on bag-specific information.

Note that $\tilde{x}_i$, whose dimension is only related to the dimension of parameter $\theta$ [31]. In other words, bags with different numbers of instances can be represented as new feature vectors with a consistent dimension. It is beneficial to algorithm implementation and popularization. More details can be found in the appendix.

## 3.2 Coding-based Ensemble for Imbalance

After obtaining $\tilde{x}_i$, imbalance is mainly reflected in two aspects. On the one hand, a specific label class may contain a large number of negative samples and a few positive samples. On the other hand, the number of samples that different label classes have varies greatly. In this case, traditional MIML methods pay more attention to majority classes while ignoring important information from minority ones. It results in biased models, which may mistakenly classify minority-class samples as majority-class samples during prediction. To tackle these problems, our IMIMLC constructs the coding-based ensemble model by three stages: encoding, training base classifiers, and decoding. Specifically, to rectify mistakes made by the interference of the label imbalance issue, IMIMLC trains multiple base classifiers on randomly partitioned small-scale data blocks based on error-correcting codes that correspond to subtasks randomly selected from the decomposed MLL task.

**Encoding:** Denote $Q$ as the transformed MLL task with $C$ labels, which can be decomposed into several equally sized subtasks, each of them containing $k$ labels, where $k \leq C$. Subtask set $Q^k$, which involved $k$ labels of interest, consists of $\mathbb{C}_C^k$ subtasks. IMIMLC randomly selects $d$ subtasks $\{w_i|i = 1, 2, \cdots, d\}$ from $Q^k$ without replacement and encodes them. Encoding follows that the one-vs-all encoding strategy (OVA) [28] is applied to selected $k$ labels, and "0" is assigned to labels that are not selected. Therefore, we obtain the encoding matrix $\mathbf{Z}^{(i)} \in \{-1, 0, +1\}^{C \times k}, i = 1, 2, \cdots, d$ of subtasks. Finally, the whole training encoding matrix $\mathbf{Z} \in \{-1, 0, +1\}^{C \times kd}$ is concatenated by $\mathbf{Z}^{(i)}$ with linking codewords from the same label.

**Training:** As shown in Fig. 2 (b), IMIMLC treats each column of $\mathbf{Z}$ as the label code and constructs the binary classifier $h_j(\cdot)$ based on each row of $\mathbf{Z}$. It means that values "0", "1", and "-1" in $\mathbf{Z}$ not only represent codewords for labels but also indicate which bags need to be involved in the training of $h_j(\cdot)$ and play the role of positive or negative categories in this process. In detail, if the label is encoded by "+1", bags related to that label will be treated as positive samples for $h_j(\cdot)$. If the label is encoded by "-1", bags related to that label and not related to any label encoded by "+1" will be treated as negative samples for $h_j(\cdot)$. If the label is encoded by "0", bags only related to that label will be excluded from training the base classifier $h_j(\cdot)$.

It is noteworthy that training bags of $h_j(\cdot)$ are selected by OVA-based coding and relationships among labels. As the number of subtasks increases, label correlation is more fully utilized, and model performance will be improved. Besides, the imbalance problem in small-scale data blocks may not be so serious. Different binary base classifiers are trained on different data blocks, which enhances the diversity of classifiers and recognition abilities for minority classes.

**Decoding:** Conducting all base classifiers on $\tilde{x}_i$, we can get a code $z_i$ composed of corresponding codewords $c_i^1, c_i^2, \cdots, c_i^{kd}$, which are binary classification results. The matching of $\tilde{x}_i$ and the label set relies on the hamming distance between $z_i$ and all label codes. However, elements of the $l$-th label code $z^l$ may contain "0", as the $l$-th label is irrelevant to the training of the corresponding base classifier, and that pair of codewords will certainly be different, resulting in a larger hamming distance. Such meaningless codeword pairs should be excluded to obtain more accurate and reliable values. Therefore, the valid hamming distance between two codewords can be defined as

$$D_V(z^l, z_i) = D(z^l, z_i) - \sum_{s=1}^{kd} \mathbf{1}_{c_l^s}, \qquad (2)$$

where $\mathbf{1}_{c_i^s} = 1$ when $c_l^s = 0$, otherwise $\mathbf{1}_{c_i^s} = 0$.

Typically, if $D_V$ between $z_i$ and $z^l$ is less than the threshold $\eta_D$, $\mathbf{B}_i$ can be matched with the $l$-th label. As we have analyzed, the matching result is determined by multiple binary classification results. When the imbalanced distribution interferes the learning of the binary classifier, the ensemble model can correct this error and output more reliable results. However, because different labels participate in the training of different numbers of base classifiers, their meaningful codeword lengths are different, as the quantity of meaningful codeword pairs is determined by the number of base classifiers that label participates in. It is unfair to use a fixed threshold $\eta_D$ to cover all label judgments. Hence, we match the bag and label semantics by $t_l^i$,

$$t_l^i = \frac{D_V(z^l, z_i)}{kd - \sum_{s=1}^{kd} I_{c_l^s}}, \qquad (3)$$

which is related to the number of base classifiers whose training the $l$-th label participates in. $1 - t_l^i$ can be regarded as the matching degree between the $i$-th bag and the $l$-th label semantics. Finally, $p_{z_i} = \mathbf{1}_C - t^i$ can be viewed as a new feature representation that contains semantic information, or the soft label set of $\mathbf{B}_i$.

## 3.3 Adaptive Threshold Method with Classifiers

In multi-label scenarios, it is not easy to obtain the precise tags of the object based on soft labels. In imbalanced MIML tasks, it is more challenging. The multiple semantics of the bag make it correspond to different numbers of labels. With the increasing dimensionality of the label space, differences among soft labels are tiny. Small changes can lead to misjudgments. Besides, in label imbalance cases, the fixed threshold adopted by traditional methods may incorrectly label the bag as majority classes with high prior probabilities. To obtain more reliable predictions, we adaptively learn classification thresholds for multiple label classes.

Concretely, we decompose this labeling task into multiple classification tasks, establishing a binary classifier for each label. The process of adapting the threshold is equivalent to maximizing the margin between two half spaces, the "larger than" one and the "less than" one. Since the input $\mathbf{X}_p = [\boldsymbol{p}_{z_1}, \boldsymbol{p}_{z_2}, \cdots, \boldsymbol{p}_{z_M}]$ of classifiers contains semantics and label correlation has been considered in the training of the ensemble model, it is acceptable to perform decomposition here. Decomposition is beneficial for adapting thresholds for different labels more pertinently without being affected by majority class priors or label dependence. Hence, its objective can be formulated as the following constraint optimization problem

$$
\begin{cases}
\min_{\boldsymbol{\omega}_l, b_l, \xi_i} \dfrac{1}{2}\boldsymbol{\omega}_l^T \boldsymbol{\omega}_l + \lambda \sum_{i=1}^{M} \xi_i \\
\text{s.t. } y_i^l(\boldsymbol{\omega}_l^T \phi(\boldsymbol{p}_{z_i}) + b_l) \geq 1 - \xi_i, \ \xi_i \geq 0,
\end{cases}
\tag{4}
$$

where $1 \leq i \leq M$, $\phi(\cdot)$ maps $\boldsymbol{p}_{z_i}$ into a higher-dimensional space, and $\lambda$ is the non-negative regularization hyper-parameter. The problem in Eq. (4) is a classical convex quadratic problem with inequality constraints. Motivated by [29], it can be equally transformed into the following unconstrained optimization problem by imposing the Lagrangian multipliers $\boldsymbol{\alpha}, \boldsymbol{\beta}$

$$
\begin{aligned}
L(\boldsymbol{\omega}_l, b_l, \xi_i, \alpha_i, \beta_i) &= \frac{1}{2}\boldsymbol{\omega}_l^T \boldsymbol{\omega}_l + \lambda \sum_{i=1}^{M} \xi_i \\
&- \sum_{i=1}^{M} \alpha_i(y_i^l(\boldsymbol{\omega}_l^T \phi(\boldsymbol{p}_{z_i}) + b_l) - 1 + \xi_i) - \sum_{i=1}^{M} \beta_i \xi_i,
\end{aligned}
\tag{5}
$$

i.e., $\min_{\boldsymbol{\omega}_l, b_l, \xi_i} \max_{\alpha_i, \beta_i} L$, which is further equivalent to $\max_{\alpha_i, \beta_i} \min_{\boldsymbol{\omega}_l, b_l, \xi_i} L$ based on Karush-Kuhn-Tucker (KKT) [12, 14]. Obeying the KKT conditions, the dual problem of Eq. (4) can be equally formulated as

$$
\begin{cases}
\min_{\boldsymbol{\alpha}} \dfrac{1}{2} \sum_{i=1}^{M} \sum_{j=1}^{M} \alpha_i \alpha_j y_i^l y_j^l \left( \phi(\boldsymbol{p}_{z_i})^T \phi(\boldsymbol{p}_{z_j}) \right) - \sum_{i=1}^{M} \alpha_i \\
\text{s.t. } \sum_{i=1}^{M} \alpha_i y_i^l = 0, \ 0 \leq \alpha_i \leq U, \ i, j = 1, 2, \cdots, M.
\end{cases}
\tag{6}
$$

Many algorithms related to quadratic programming can be used to deal with the problem in Eq. (6). We adopt the sequential minimal optimization (SMO) algorithm, which is faster and more efficient [27], to get the optimal solution $\boldsymbol{\alpha}^*$. According to the primal-dual relationship and KKT, optimal $\boldsymbol{\omega}_l^*$ satisfies

$$
\boldsymbol{\omega}_l^* = \sum_{i=1}^{M} \alpha_i^* y_i^l \phi(\boldsymbol{p}_{z_i}).
\tag{7}
$$

Finally, the adaptive threshold for the $l$-th label $\eta_A^l$ can be obtained by SMO [27] or calculated as

$$
\eta_A^l = \frac{1}{M} \sum_{i=1}^{M} y_i^l - \boldsymbol{\omega}_l^{*T} \phi(\boldsymbol{p}_{z_i}).
\tag{8}
$$

## 3.4 Testing and Predicting

For a new testing bag $\hat{\mathbf{B}}$, its structure information from instances $\{\hat{\boldsymbol{x}}^1, \hat{\boldsymbol{x}}^2, \ldots, \hat{\boldsymbol{x}}^{\hat{n}}\}$ is embedded in $\tilde{\hat{\boldsymbol{x}}}$ by Eq. (1). Then $\tilde{\hat{\boldsymbol{x}}}$ will be updated as a new vector $\boldsymbol{p}_{\hat{z}}$ with semantic annotation via the coding-based ensemble model proposed in Subsection 3.2. According to trained classifiers $\{\boldsymbol{\omega}_l^* | l = 1, 2, \cdots, C\}$ on $\boldsymbol{p}_{z_i}$ by Eq. (7) and Eq. (8), we can compute the prediction score $\boldsymbol{\omega}_l^{*T} \phi(\boldsymbol{p}_{\hat{z}})$ and $\eta_A^l$. The comparison of $\boldsymbol{\omega}_l^{*T} \phi(\boldsymbol{p}_{\hat{z}})$ and $\eta_A^l$ determines the positive/negative output under the classifier for the $l$-th label. According to T-criterion [1], if $\boldsymbol{p}_{\hat{z}}$ is judged as positive by the binary classifier, $\hat{\mathbf{B}}$ is marked by the corresponding label; if all classifiers judge $\boldsymbol{p}_{\hat{z}}$ as negative, $\hat{\mathbf{B}}$ will be marked by the label that has the fewest negative marks. In other words, the hard label set of testing bag $\hat{\mathbf{B}}$ is

$$
\hat{\mathbf{Y}} = \{l | \boldsymbol{\omega}_l^{*T} \phi(\boldsymbol{p}_{\hat{z}}) \geq -\eta_A^l, l \in \mathcal{Y}\} \cup \left\{ \arg\min_{l \in \mathcal{Y}} \sum_{i=1}^{M} \mathbf{1}_{-\eta_A^l}\left(\boldsymbol{\omega}_l^{*T}\phi(\boldsymbol{p}_{z_i})\right) \right\},
$$

where $\mathbf{1}_a(x)$ is the indicator function, whose value is 1 when $x < a$.

## 3.5 Computational Complexity

The computational complexity of IMIMLC can be analyzed in two parts: (1) bag vector representation based on instances, and (2) imbalanced learning based on the coding ensemble and adaptive thresholds. We estimate the complexity as $O(N_I G d_2(T_1 + 1) + kd(T_2 N_s^3 + N_s d_1 + 3MC) + CN_s^3)$. More analyses are in the appendix.

## 4 Experiments

### 4.1 Datasets

We conducted experiments on six public MIML datasets, including MIML-image, MIML-text, HJA Bird Song, MSRC v2, Letter Carroll, and Isoform Gene Data. Their brief information is reported in Table 1. Specific information about datasets is in the appendix.

**Table 1: A brief description of datasets.**

| Name | #Labels | #Instances | Dim | #Bags | AIR |
|------|---------|-----------|-----|-------|-----|
| IMG  | 5  | 18000 | 15  | 2000  | 0.3309 |
| TEX  | 7  | 7119  | 243 | 2000  | 0.2270 |
| HBS  | 13 | 10232 | 38  | 548   | 0.2127 |
| MSRC | 23 | 1758  | 48  | 591   | 0.1368 |
| LC   | 26 | 717   | 16  | 166   | 0.2066 |
| IGD  | 94 | 59297 | 254 | 11946 | 0.0080 |

Note that different datasets have different extents of label imbalance issues. To measure the level of label imbalance in different datasets, inspired by [19], we adopt the average imbalance ratio (AIR), which is formulated as

$$
\text{AIR} = \frac{1}{C} \sum_{i=1}^{C} \frac{|N_{i+}|}{|N_{i-}|},
\tag{9}
$$

**Table 2: Performance comparisons of our IMIMLC on various datasets. The bold red font indicates the best performance, and the blue font indicates the second-best performance. ●/ ⊙ /○ means that IMIMLC is better/tied/worse than other methods. (pairwise single-tailed t-test at 95% confidence level). "N/A" indicates that the result is not available within 100 hours.**

| Dataset | Metrics | IMIMLC | MIMLSVM | MIMLSVMmi | MIMLNN | KISAR | MIMLwel | CM2AL | MK-EnMIMLNN | MIML-LLMC |
|---|---|---|---|---|---|---|---|---|---|---|
| IMG $k=3$ $d=10$ | AUC (↑) | **.837±.004** | .801±.006● | .738±.002● | .794±.008● | .834±.007⊙ | .750±.007● | .836±.007⊙ | .809±.009● | .832±.006● |
| | F1 (↑) | **.639±.011** | .583±.008● | .393±.009● | .491±.009● | .510±.008● | .529±.010● | .615±.016● | .447±.012● | .584±.015● |
| | ACC (↑) | .827±.006 | .807±.004● | .807±.002● | .807±.004● | .834±.002⊙ | .654±.008● | .828±.01⊙ | .807±.005● | **.835±.003○** |
| | HL (↓) | .173±.006 | .193±.004● | .193±.002● | .193±.004● | .166±.002⊙ | .346±.008● | .172±0.01⊙ | .193±.005● | **.165±.003○** |
| | 0-1 (↓) | .306±.015 | .040±.127○ | .433±.015● | .080±.103● | **.040±.084○** | .380±.199● | .315±0.02● | .160±.158○ | .285±.012○ |
| | RL (↓) | .189±.005 | .200±.006● | .262±.002● | .206±.008● | .166±.007○ | .250±.007● | .164±0.01○ | .191±.000⊙ | **.162±.014○** |
| TEX $k=4$ $d=21$ | AUC (↑) | .980±.001 | .944±.005● | .979±.002⊙ | .598±.131● | .977±.002● | .920±.003● | **.983±.005●** | .970±.003● | .972±.001● |
| | F1 (↑) | **.894±.006** | .821±.015● | .812±.005● | .743±.006● | .808±.005● | .746±.008● | .844±.021● | .729±.010● | .882±.004● |
| | ACC (↑) | **.976±.001** | .943±.004● | .945±.001● | .950±.001● | .958±.001● | .870±.007● | .965±.003● | .943±.001● | .973±.001● |
| | HL (↓) | **.024±.001** | .057±.004● | .055±.001● | .050±.001● | .042±.001● | .130±.007● | .035±.003● | .057±.001● | .027±.001● |
| | 0-1 (↓) | .058±.005 | .157±.045● | .074±.006● | .686±.259● | **.000±.000○** | .100±.096● | .073±.010● | .014±.045○ | .048±.002○ |
| | RL (↓) | .020±.001 | .056±.005● | .022±.002● | .031±.002● | .023±.002● | .080±.003● | .017±.005○ | .031±.003● | **.016±.001○** |
| HBS $k=4$ $d=52$ | AUC (↑) | **.974±.001** | .964±.005● | .820±.016● | .966±.005● | .788±.085● | .888±.004● | .901±.025● | .966±.004● | .523±.010● |
| | F1 (↑) | **.842±.012** | .828±.007● | .631±.010● | .788±.034● | .569±.177● | .572±.013● | .010±.007● | .818±.007● | .092±.022● |
| | ACC (↑) | **.975±.001** | .939±.003● | .916±.003● | .941±.006● | .600±.018● | .803±.009● | .848±.003● | .940±.003● | .836±.001● |
| | HL (↓) | **.025±.001** | .061±.003● | .084±.003● | .059±.006● | .400±.018● | .197±.009● | .152±.003● | .060±.003● | .164±.001● |
| | 0-1 (↓) | **.022±.004** | .054±.063⊙ | .095±.012● | .146±.044● | .500±.155● | .454±.092● | .641±.036● | .131±.052● | .688±.030● |
| | RL (↓) | **.009±.001** | .036±.005● | .233±.018● | .034±.005● | .227±.085● | .112±.004● | .092±.019● | .034±.004● | .530±.035● |
| MSRC $k=4$ $d=69$ | AUC (↑) | **.926±.003** | .914±.006● | .802±.016● | .891±.005● | .629±.012● | .891±.009● | .887±.009● | .925±.005⊙ | .872±.005● |
| | F1 (↑) | **.705±.007** | .550±.023● | .422±.015● | .394±.019● | .374±.019● | .507±.021● | .407±.028● | .538±.021● | .384±.013● |
| | ACC (↑) | **.958±.001** | .928±.003● | .918±.002● | .923±.002● | .263±.003● | .867±.003● | .938±.005● | .932±.002● | .930±.001● |
| | HL (↓) | **.042±.001** | .072±.003● | .082±.002● | .077±.002● | .737±.003● | .133±.003● | .062±.005● | .068±.002● | .070±.001● |
| | 0-1 (↓) | **.131±.005** | .252±.061● | .301±.016● | .313±.061● | .674±.023● | .552±.080● | .327±.183● | .187±.062● | .208±.006● |
| | RL (↓) | **.050±.001** | .086±.006● | .179±.008● | .110±.005● | .530±.012● | .109±.009● | .097±.025● | .075±.005● | .090±.003● |
| LC $k=3$ $d=104$ | AUC (↑) | **.919±.005** | .717±.011● | .579±.022● | .824±.013● | .629±.015● | .841±.007● | .538±.027● | .845±.008● | .647±.014● |
| | F1 (↑) | **.777±.022** | .262±.028● | .373±.023● | .276±.027● | .208±.027● | .538±.020● | .139±.008● | .270±.024● | .146±.023● |
| | ACC (↑) | **.956±.003** | .857±.005● | .883±.008● | .861±.003● | .172±.010● | .803±.009● | .852±.008● | .865±.004● | .871±.004● |
| | HL (↓) | **.044±.003** | .143±.005● | .117±.008● | .139±.003● | .828±.010● | .197±.009● | .148±.008● | .135±.004● | .129±.004● |
| | 0-1 (↓) | **.071±.012** | .727±.105● | .106±.026● | .581±.097● | .827±.027● | .485±.098● | .994±.012● | .608±.035● | .273±.030● |
| | RL (↓) | **.048±.003** | .283±.011● | .343±.032● | .176±.013● | .684±.018● | .159±.007● | .364±.029● | .155±.008● | .196±.009● |
| IGD $k=4$ $d=376$ | AUC (↑) | **.574±.004** | .254±.000● | .562±.006● | .219±.002● | .208±.002● | .215±.003● | N/A | .257±.001● | .434±.004● |
| | F1 (↑) | .008±.001 | .012±.000⊙ | .007±.001○ | .005±.001● | **.020±.000○** | .001±.000● | N/A | .000±.000● | .002±.000● |
| | ACC (↑) | .981±.000 | .982±.000● | .992±.000● | .991±.000● | .007±.000● | .991±.000● | N/A | .992±.000○ | **.992±.000○** |
| | HL (↓) | .019±.000 | .018±.000● | .008±.000● | .009±.000○ | .993±.000● | .009±.000● | N/A | .008±.000● | **.008±.000○** |
| | 0-1 (↓) | .960±.002 | .992±.005● | .966±.003● | .962±.016⊙ | 1.00±.000● | **.959±.015⊙** | N/A | .977±.015● | .976±.001● |
| | RL (↓) | **.119±.001** | .326±.001● | .438±.006● | .422±.006● | .468±.004● | .386±.008● | N/A | .317±.002● | .156±.002● |
| win/tie/loss | | — — | 42/1/5 | 43/2/3 | 43/1/4 | 38/1/9 | 42/4/2 | 31/7/2 | 41/3/4 | 39/1/8 |

where $|N_{i+}|$ and $|N_{i-}|$ represent the quantity of minority and majority class samples under the $i$-th label, respectively. AIR ranges from 0 to 1, and the more balanced the label distribution, the closer this value is to 1, otherwise, to 0. AIRs of datasets are presented in the last column of Table 1.

## 4.2 Experiment Settings

To showcase the performance of our IMIMLC, several representatives of MIML algorithms were selected for comparison. MIMLSVMmi [48] degenerates the original MIML task into an MIL task to learn the instance-level classifier. MIMLSVM [43], MIMLNN [48], MK-EnMIMLNN [18], KISAR [16], and MIMLwel [42] convert original MIML tasks into MLL tasks to learn bag-level classifiers. CM2AL [35] and MIML-LLMC [41] investigate the underlying relationships among instances, bags, and labels to learn joint classifiers. Among them, CM2AL, MK-EnMIMLNN, and MIML-LLMC are state-of-the-art algorithms. To comprehensively quantify the performance of all algorithms, we adopt six evaluation metrics, which are commonly used and listed as follows:

- **Accuracy (ACC, ↑):** The proportion of correctly predicted labels for the testing data.
- **Area Under the Curve (AUC, ↑):** The average area under the Receiver Operating Characteristics (ROC) curve.

- **F1 score (F1, ↑):** The harmonic average of precision and recall for classification.
- **Hamming Loss (HL, ↓):** The proportion of mislabeled predictions, which is the dual metric of accuracy.
- **One-Error (0-1, ↓):** The proportion of samples where the most likely label is not the true label.
- **Ranking Loss (RL, ↓):** The proportion of cases where the false soft labels are ranked higher than the true soft labels.

## 4.3 Results and Analysis

To ensure a fair comparison of classification performance, experiments were conducted with the optimal parameters of all algorithms. For each dataset, 60% of its samples were selected as the training set, 20% as the validation set, and the remaining 20% as the testing set. To eliminate the effect of random factors, experiments were independently repeated 20 times. The average and standard deviation of AUC, F1-score, accuracy, hamming loss, one-error, and ranking loss on all datasets are recorded in Table 2. As seen from the results in Table 2, we have several observations.

(1) IMIMLC outperforms other algorithms in most cases. For example, on the LC dataset, IMIMLC performed approximately 7% better than the second-best MK-EnMIMLNN in

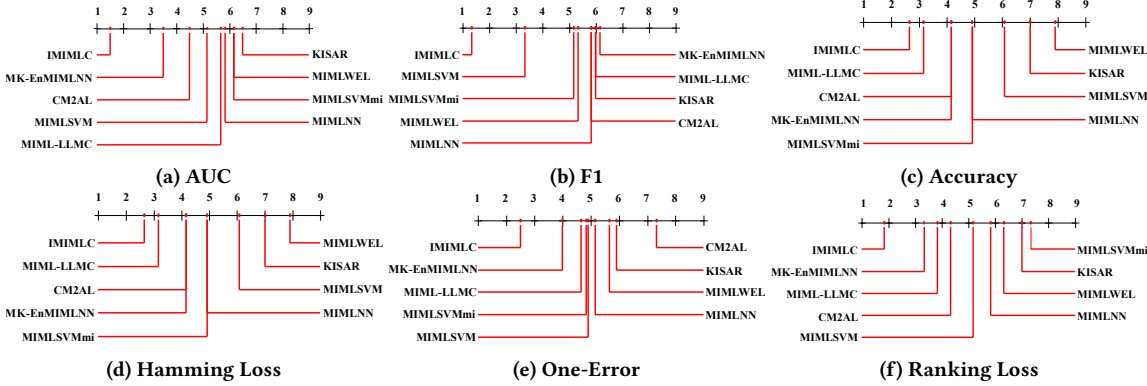

**Figure 3: Comparisons of IMIMLC with eight algorithms by the Nemenyi test in the non-parametric statistical sense.**

evaluation based on AUC. In terms of F1, IMIMLC even out-performed second-best MIMLSVMwel by approximately 24%, which was a significant improvement. The reason may be that IMIMLC has mitigated the imbalance issue with the coding-based ensemble and an adaptive threshold strategy.

(2) Six evaluation metrics, AUC, F1, ACC, 0-1, RL, and HL, are adopted to evaluate the performance of all MIML methods from different aspects. In different tasks and scenarios, the ranks of their classification performance based on different metrics are different. For example, on the TEX dataset, the AUC of our IMIMLC was slightly inferior to that of CM2AL, while its F1 score was significantly better than others. Thus, with imbalanced data, we should adopt multiple evaluation metrics to verify the effectiveness of algorithms.

(3) Comparison methods KISAR and MIML-LLMC sometimes performed more competitively than IMIMLC on the IMG dataset. However, these algorithms have only considered one aspect of our scenario setting without taking into account the issue of label distribution imbalance, which makes them less effective than IMIMLC in more experiments.

To deeply evaluate the performance of all comparison methods, we conducted statistical comparisons based on parametric and non-parametric tests. For the parametric test, t-tests evaluate the statistical differences between two specific algorithms on individual metrics in each dataset. For the non-parametric test, the Nemenyi tests compare performance based on the average ranks of algorithms in all datasets. Their results are presented in Table 2 and Fig. 3, from which we have the following observations.

(1) The t-test results in the last row of Table 2 demonstrate that the performance of IMIMLC is still better than others in most cases. In paired competitions with the t-test, the win rate of IMIMLC was at least 77.50%, sometimes even 89.58%. Besides, different datasets have different extents of label imbalance issues. Even in datasets with heavy imbalances, IMIMLC can still play to its strengths and win in t-tests.

(2) Fig. 3 shows critical difference diagrams of Nemenyi tests based on all metrics. In each subfigure, the average ranks of algorithms are marked along the axis, with lower ranks to the left. IMIMLC achieved the lowest average rank against

other comparison approaches in terms of all metrics. It has been proven that IMIMLC is better than others. Besides, there are some specials in these subfigures. In Fig. 3c, although IMIMLC is at the top of the ranking, its rank exceeds "2" and approaches "3". It reminds us that in imbalanced datasets, evaluations based on accuracy are unstable. It is necessary to construct a comprehensive evaluation system.

## 4.4 Ablation Study

To better verify the effectiveness of our IMIMLC, we conducted ablation studies to investigate the effect of each component. Firstly, we validate the effectiveness of feature embedding based on the fisher vector and fusion coding-based ensemble. Specifically, in stage 1, we replaced our method with a classical method, which extracts features by calculating the average of instance feature values, to validate the effectiveness of the feature embedding approach adopted by IMIMLC. In stage 2, we replaced our strategy with MESA, which adopts decision trees (DTs) and SVMs as base classifiers, to validate the effectiveness of the special ensemble model adopted by IMIMLC for addressing the issue of imbalanced label distribution. Thereby, six variants in the ablation experiment were formed.

Results on the IMG and MSRC datasets are presented in Table 3. In stage 1, feature embedding improves model classification ability to some extent. However, only conducting this in stage 1 does not significantly improve model performance. In stage 2, the adoption of the fusion coding-based ensemble effectively alleviates the imbalance problem and results in excellent performance in most metrics. However, only using our strategy in stage 2, it competed with MESA in terms of F1 score. When both stages used strategies proposed in IMIMLC, classification performance had an overall improvement of approximately 5%-10%, even some metrics increased by 20% on the MSRC dataset, which means that the ability of the model to tackle imbalanced MIML tasks is dramatically enhanced.

Besides, to further validate the effectiveness of the adaptive threshold method with classifiers in IMIMLC, we conducted experiments based on fixed thresholds of 0.5, 0.6, 0.7, 0.8, and 0.9 and adaptive thresholds. Experimental results based on three metrics, which are mainly influenced by threshold-based hard labels, on the TEX dataset are recorded in Table 4, and more results on other datasets can be found in the appendix. It is obvious that our

**Table 3: Comparisons of different feature embedding algorithms and imbalanced learning algorithms on IMG and MSRC datasets. "✓" indicates our strategy, and "✗" indicates others.**

| | Stage 1 | Stage 2 | ACC (↑) | AUC (↑) | F1 (↑) | HL (↓) | 0-1 (↓) | RL (↓) |
|---|---|---|---|---|---|---|---|---|
| | ✗ | ✗-DT | .761±.005 | .775±.006 | .551±.007 | .239±.005 | .410±.009 | .268±.008 |
| | ✗ | ✗-SVM | .771±.026 | .834±.004 | .614±.014 | .229±.026 | .414±.051 | .201±.017 |
| | ✗ | ✓ | .812±.005 | .821±.003 | .591±.010 | .188±.005 | .327±.009 | .199±.005 |
| IMG | ✓ | ✗-DT | .762±.004 | .775±.005 | .560±.009 | .238±.004 | .406±.017 | .263±.006 |
| | ✓ | ✗-SVM | .781±.011 | .835±.004 | .614±.008 | .219±.011 | .388±.027 | .191±.005 |
| | ✓ | ✓ | **.827±.006** | **.837±.004** | **.639±.011** | **.173±.006** | **.306±.015** | **.189±.005** |
| | ✗ | ✗-DT | .856±.012 | .886±.011 | .521±.012 | .144±.012 | .250±.023 | .122±.010 |
| | ✗ | ✗-SVM | .797±.012 | .875±.006 | .461±.011 | .203±.012 | .407±.030 | .119±.008 |
| | ✗ | ✓ | .926±.003 | .887±.009 | .403±.025 | .074±.003 | .285±.017 | .095±.005 |
| MSRC | ✓ | ✗-DT | .883±.004 | .898±.006 | .558±.009 | .117±.004 | .203±.013 | .113±.004 |
| | ✓ | ✗-SVM | .875±.008 | .923±.006 | .605±.011 | .126±.008 | .181±.020 | .069±.002 |
| | ✓ | ✓ | **.958±.001** | **.926±.003** | **.705±.007** | **.042±.001** | **.131±.005** | **.050±.001** |

adaptive threshold strategy is more beneficial for improving model classification performance than the fixed threshold method.

**Table 4: Comparisons of classification performance with different thresholds on the TEX datasets.**

| $\eta$ \ Metrics | HL (↓) | ACC (↑) | F1 (↑) |
|---|---|---|---|
| $\eta_p = 0.5$ | .775±.002 | .225±.002 | .284±.001 |
| $\eta_p = 0.6$ | .291±.005 | .709±.005 | .500±.006 |
| $\eta_p = 0.7$ | .119±.001 | .881±.001 | .758±.005 |
| $\eta_p = 0.8$ | .074±.002 | .926±.002 | .826±.008 |
| $\eta_p = 0.9$ | .051±.001 | .949±.001 | .702±.005 |
| $\eta_A$ | **.024±.001** | **.976±.001** | **.894±.006** |

## 4.5 Hyper-parameter Sensitivity Analysis

IMIMLC involves two hyper-parameters: subset size $k$ and selected subtask number $d$. To enhance sample diversity and alleviate label imbalance issues, the number of subtasks in $Q^k$, $\mathbb{C}_C^k$, should be maximized. Similar to [32], $k$ would not be too large, and it was controlled to be within $\{2, 3, 4, 5, 6\}$. Besides, $d$ shouldn't be too small because small-scale subtasks could not cover sufficient labels. Therefore, $d$ was controlled to be within $\{C, 2C, 3C, 4C\}$. To better declare the effect of these hyper-parameters on the performance of IMIMLC, we report AUCs based on different parameter combinations in two datasets with high label space dimensions in Fig. 4. Results based on other datasets are in the appendix.

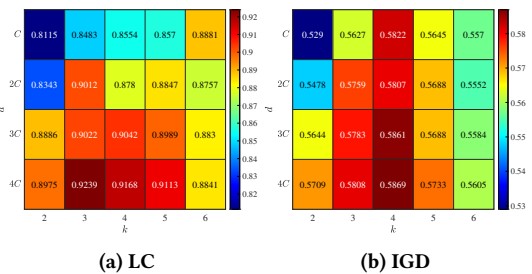

| (a) LC | (b) IGD |
|---|---|

**Figure 4: Sensitivity analysis with different $k$ and $d$.**

As presented in Fig. 4, when $k = 3$ or $k = 4$, IMIMLC demonstrates better performance, and as $d$ increases, the capability of the algorithm improves. Obviously, the optimal hyper-parameter combination is relatively easy to obtain.

## 4.6 Computational Efficiency Analysis

To further compare the computational efficiency of algorithms, we have recorded the running times of all algorithms on six datasets in Fig. 5. As shown in Fig. 5, on the IMG dataset, IMIMLC is indeed more efficient and faster. Compared to KISAR, which has the second-shortest running time, our method had a speed improvement of approximately 22.3%. Additionally, the number of labels, instances, and dimensions also affect the efficiency of the algorithm. On other datasets, IMIMLC did not perform at the fastest computing speed, but it remained at a relatively high level.

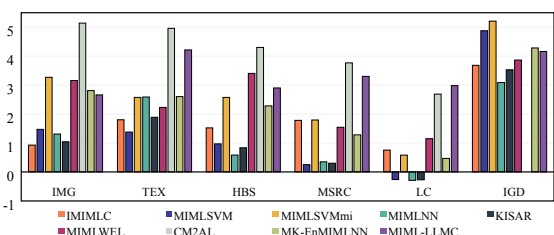

**Figure 5: Comparisons of the relative logarithmic running times of nine algorithms on six datasets. The Y-axis is scaled by log to mitigate the gap between algorithms. A missing bar indicates that the result is not available within 100 hours.**

## 5 Conclusion

We discuss the MIML task in the context of imbalanced label distribution and propose IMIMLC based on the error-correcting coding ensemble and an adaptive threshold strategy to alleviate the effect of imbalance. IMIMLC trains the ensemble model on randomly selected data blocks to enhance the diversity of base classifiers and adaptively learns its thresholds for each semantic class label to obtain more reliable predictions than the manually pre-specified schemes. Finally, extensive experimental results show the effectiveness of IMIMLC against state-of-the-art MIML methods in imbalanced label distribution scenarios. In the future, we will extend IMIMLC to solve problems in different cases, such as noisy labels, novel classes, etc [34, 46, 47]. Besides, there are many more complex nonlinear features in real-word applications. To handle such tasks, we will study the model that ensembles various types of base nonlinear classifiers, such as decision trees, neural networks, etc.

# Acknowledgments

This work was supported by the National Science Foundation of China Grant [62036013, 62376281], and the NSF for Huxiang Young Talents Program of Hunan Province under Grant [2021RC3070].

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
