# OpenReview forum: "Imbalanced Multi-instance Multi-label Learning via Coding Ensemble and Adaptive Thresholds"
_acmmm.org/ACMMM/2024/Conference — MM2024 Poster_

### Official Review · Reviewer_RKaa · 2024-05-21

**Rating:** 3
**Confidence:** 3

**Summary:**

The paper proposes a novel imbalanced multi-instance multi-label learning method named IMIMLC, which addresses the issue of label imbalance in MIML tasks. The method uses error-correcting coding ensemble and adaptive threshold strategy to handle the imbalance problem. Experimental results on various datasets demonstrate the effectiveness of IMIMLC compared to state-of-the-art approaches.

**Strengths:**

1	The paper presents extensive experimental results across various datasets, highlighting the superior effectiveness of IMIMLC in comparison to other methods.
2	The motivation and the definition of the problem in the introduction are sufficient.
3	From the point of experiment result, the performance of the IMIMLC is good.

**Limitations:**

1 From a theoretical point of view, I am not sure of the novelty of the proposal. In fact, it seems that it can be developed further by considering, for instance, Imbalance problems strategy more general than ensemble strategy.
2 The imbalanced learning methods mentioned by the author in Chapter 1 and 2.2 imbalanced learning have nothing to do with multi-label learning. The author should conduct a review and analysis of the existing imbalanced learning methods studied in multi-label learning.
3	The experimental analysis is confusing. The author designed some metrics for comparison. For the reader, he does not know which metric is beneficial.

**Suitability:**

2

---

### Official Review · Reviewer_FYGc · 2024-05-23

**Rating:** 3
**Confidence:** 3

**Summary:**

This study introduces an innovative unbalanced multi-instance multi-label learning (IMIMLC) approach, integrating error-correcting coding with adaptive thresholding. Utilizing Fisher vector-based feature embedding, it captures object structure and trains binary classifiers on compact data subsets for enhanced prediction accuracy. The method reframes threshold determination as a classifier optimization task, aiming to maximize soft label boundaries and dynamically assign thresholds per label.

**Strengths:**

(1)This article is composed in a systematic and comprehensive style, indicating that the author has cultivated proficient academic writing skills.
(2)This article rigorously tackles the challenge of label imbalance inherent in Multi-Instance Multi-Label (MIML) learning and proposes a novel methodology that synergizes coding-integrated learning with adaptive classification thresholding techniques.
(3)The experimental section is adequate and complete and persuasive.

**Limitations:**

(1)In the Training part of the Methods section, a more elaborate exposition on the integrated learning module is imperative to enhance comprehension, particularly for individuals outside the field. This entails detailing aspects such as the specific integrated learning model employed, its objective function, relevant formulas, theoretical analysis, among other critical components.
(2)The meaning of Eq4 needs to be explained in more detail, e.g. what is wl?

**Suitability:**

3

---

### Official Review · Reviewer_6qc7 · 2024-06-09

**Rating:** 5
**Confidence:** 2

**Summary:**

The paper explores the problem of imbalanced label distribution in multi-instance multi-label learning and proposes IMIMLC based on the error-correcting coding ensemble and an adaptive threshold strategy. Extensive experiments have demonstrated its outstanding performance.

**Strengths:**

1. The problem statement (solving imbalanced label distribution for MIML task) makes sense, and I believe it holds significant importance for MIML tasks.
2. The experimental analysis is comprehensive and reasonable.

**Limitations:**

The paper presents a clear and intuitive idea and I do not see critical weaknesses. However, some more analysis can be provided on where and to what extent the proposed method works better. For example:

1. Table 2 shows that the proposed method seems to achieve optimal performance across nearly all metrics. However, it is hard to tell how strong the proposed method can perform for minority classes. Is the performance improvement attributable to improvements in these minority classes? A more detailed analysis of the performance gains the proposed method offers for classes with different proportions can be provided.
2. Similarly, Table 3 presents a comparison of different feature embedding algorithms and imbalanced learning algorithms. Can you provide an analysis of their results for classes with different proportions?

**Suitability:**

3

---

### Meta-Review · Area_Chair_iv24 · 2024-07-02

**Recommendation:** Accept (Poster)
**Confidence:** 5

**Metareview:**

Reviewers provided 3 distinct views on the topic of imbalanced label distribution in multi-instance multi-label learning. I support the paper to be accepted as a poster, as this is a topic that clearly needs more discussion and exploration in our community.